# Modeling and Testing of a Composite Steel–Concrete Joint for Hybrid Girder Bridges

**DOI:** 10.3390/ma16083265

**Published:** 2023-04-21

**Authors:** Bing Shangguan, Qingtian Su, Joan R. Casas, Hang Su, Shengyun Wang, Rongxin Zhao

**Affiliations:** 1Department of Bridge Engineering, Tongji University, 1239 Siping Road, Shanghai 200092, China; shangguanbing@tongji.edu.cn (B.S.); sqt@tongji.edu.cn (Q.S.); 2010034@tongji.edu.cn (H.S.); 2232463@tongji.edu.cn (S.W.); 2Guangdong Yejian Construction Drawing Review Center Co., Ltd., Guangzhou 510060, China; 3Shanghai Engineering Research Center of High Performance Composite Bridge, Shanghai 201306, China; 4School of Civil Engineering, Department of Civil and Environmental Engineering, Technical University of Catalunya, 08034 Barcelona, Spain; 5Shanghai Research Institute of Building Sciences Co., Ltd., Shanghai 201108, China

**Keywords:** hybrid bridge, steel–concrete joint, static test, load-carrying capacity, numerical study, parametric analysis

## Abstract

A hybrid girder bridge adopts a steel segment at the mid-span of the main span of a continuous concrete girder bridge. The critical point of the hybrid solution is the transition zone, connecting the steel and concrete segments of the beam. Although many girder tests revealing the structural behavior of hybrid girders have been conducted by previous studies, few specimens took the full section of a steel–concrete joint due to the large size of prototype hybrid bridges. In this study, a static load test on a composite segment to bridge the joint between the concrete and steel parts of a hybrid bridge with full section was conducted. A finite element model replicating the tested specimen results was established through Abaqus, while parametric studies were also conducted. The test and numerical results revealed that the concrete filling in the composite solution prevented the steel flange from extensive buckling, which significantly improved the load-carrying capacity of the steel–concrete joint. Meanwhile, strengthening the interaction between the steel and concrete helps to prevent the interlayer slip and simultaneously contributes to a higher flexural stiffness. These results are an important basis for establishing a rational design scheme for the steel–concrete joint of hybrid girder bridges.

## 1. Introduction

Under the effect of a multifaceted load, a prestressed concrete (PSC) girder bridge [1,2,3,4] with a long span has common disadvantages of long-term downward deflection, web cracking and prestress loss, which have an adverse impact on the ultimate strength and serviceability limit states. Hybrid continuous bridges with PSC around the supports and steel in the mid-span area solve this problem by allocating the materials along the bridge where they can perform better (as presented in Figure 1). Steel–concrete hybrid girder requires a composite steel–concrete transition joint to connect the steel and concrete parts along the longitudinal direction. At present, hybrid girders are used in beam-like bridges [5,6,7,8], cable-stayed bridges [9,10,11] and suspension bridges [12,13]. Hybrid girder bridge adopts a steel–concrete composite segment at mid-span of a continuous span, which not only inherits the advantages of easy construction and good economy of the prestressed concrete, but also reduces the self-weight due to the use of steel around the mid-span. In addition, the high weight of the concrete in the side spans contributes to smaller deflection and internal forces in the main span, leading to lower cost and feasibility of the construction of longer spans.

Since the structural form of hybrid girders was proposed, a large number of studies have been carried out to study a better connection type between steel and concrete parts to facilitate the transition of stresses from one to the other. Kim tested three connection types on small-scale steel–PSC hybrid girders, and a connection consisting of parallel perfobond ribs was recommended [14,15]. Chen conducted numerical and analytical analysis on the joint section of hybrid girders and verified the smooth stress transition within the joint section [16]. Xing carried out a numerical study on the effectiveness of shear connectors in the joint section of hybrid girders [17]. The numerical results showed how the shear connectors contributed to a reliable connection between steel and concrete, while the prestressed system is also conducive to strengthen the connection. Jiang conducted model tests on a pressure-resistant shear-transferred steel–concrete joint of hybrid girders and showed a higher bearing capacity of steel–concrete joints [18]. Qin carried out research on the optimal position of the steel–concrete joint section of the hybrid girder bridge through a parametric sensitivity analysis [19]. Zhang proposed a new type of connection form to facilitate the constructability of the steel-to-concrete connection and confirmed its good seismic behavior in precast hybrid steel–concrete girders [20]. He deduced a relationship between the optimal length of the steel–concrete connection and the length ratio of side span to main span [21], which was proved to be an efficient method for the preliminary design of hybrid girder bridges.

The analysis of steel-to-concrete connections refers not only to hybrid beams but also to classical and non-standard composite beams. Su revealed the influence of the steel–concrete interaction on the overall mechanical performance of composite girders [22,23,24]. Tesser presented 24 lab tests on composite steel trusses and concrete beams with an inferior precast concrete base [25], while the main features of the composite steel trusses and concrete beams are quantitatively and qualitatively discussed. Colajanni investigated the failure modes and the stress transfer mechanism between concrete and embedded steel elements of semi-precast hybrid-steel-trussed concrete beams [26,27]. The influence of steel–concrete interactions on the failure modes and seismic damping capacity were also discussed.

Although many girder tests revealing the structural behavior of hybrid bridges have been conducted, few specimens took the full section of the steel–concrete joint due to the large dimensions of a prototype hybrid bridge. Tests results from a partial section of a girder may reveal the local stress conditions of the steel–concrete joint in the hybrid girder, but its overall structural behavior cannot be obtained directly. When laboratory tests cannot be performed, the size effect on the structural performance were typically evaluated through numerical models. The present research aims to experimentally and numerically investigate the overall mechanical behavior of the steel–concrete joint in hybrid girders. A simply supported composite girder taking the full section of a prototype bridge was tested for a detailed structural performance investigation. A finite element model of the specimen was established through Abaqus and calibrated with the test results. Then, parametric studies were also conducted to derive rational design criteria of the joint section. The objective of the present study is to take the first step in investigating the overall flexural behavior of a steel–concrete joint in hybrid girders and help establish appropriate design criteria for the transition zone and the joint section. The results obtained here need to be complemented with further tests mainly focused on the local mechanisms of response of the joint.

## 2. Tests Description

### 2.1. Specimen Design

The steel–concrete joint section in the longitudinal direction proposed in this study is based on a composite steel–concrete solution in the transverse direction. Therefore, a simply supported composite girder was manufactured and tested. The specimen was inspired on the transition segment of the Anhai Bay bridge, which is a key link in the construction of China’s first cross-sea high-speed railway. The Anhai bridge, located in Quanzhou City, is a three-span hybrid-beam rigid-frame bridge with a span arrangement of 135 + 300 + 135 m. In the main span, the 103 m long steel box girder is connected monolithically to the concrete box girder by a transition segment (5 m long) at each end with a cross-section depth of approximately 5.9 m (as shown in Figure 2). The bridge has become the world’s second largest span hybrid-beam rigid-frame bridge.

Figure 3 shows the configuration details of the specimen, which was a 1:8 scale model of the Anhai Bay Bridge. The girder test aimed to obtain the sectional bending capacity of the specimen. Thus, the geometric parameters of the section of the specimen were determined as one-eighth of the prototype bridge. As the test model is a small-scale model, the two webs of the box section will be too thin for a proper manufacturing. Thus, the specimen combined the two webs together, while the section was regarded as a steel girder filled with concrete in such a way that the specimen section had the same flexural rigidity as the original section in the bridge. The thickness of all steel plates was 3 mm. The composite girder height was 750 mm, and the width of the top and bottom flanges was 860 mm and 496 mm, respectively (as presented in Figure 3a). Five layers of longitudinal reinforcing bars and two layers of transversal reinforcing bars were included in the web and flanges, respectively. The spacing of the longitudinal and transversal reinforcing bars was 150 mm and 50 mm, respectively, while their diameter was 12 mm. The full girder length was 8 m, including a 250 mm free extension at each end (as presented in Figure 3b). Holes were reserved in advance in the horizontal and vertical stiffeners of the steel beam to facilitate the posterior placement of the rebars. Photos during the specimen construction are shown in Figure 4. The structural form of a perfobond rib was adopted for the steel–concrete connection (as is presented in Figure 4b,c). Figure 4a–c show the fabrication of the steel girder and positioning of the reinforcing bars. Figure 4d,e show the concrete casting.

### 2.2. Material Properties

The compressive strength of the concrete was 50.4 MPa as obtained from three 150 mm cubic probes tested on the 28th day after casting [28]. Six steel plate specimens and reinforcing bars were also tested. Results showed that the tensile yielding and ultimate strengths of the steel plate was 393 MPa and 573 MPa, respectively. The reinforcing bars exhibited no evident yielding point, while the ultimate strength was 613 MPa.

### 2.3. Loading and Monitoring Setup

The specimen was loaded at mid-span using a 2000 kN capacity hydraulic jack (as presented in Figure 5). The loading protocol included a 50 kN preload and monotonic loading up to the ultimate state. Figure 6 shows the positions of the sensors, including deflection sensors and strain gauges. The deflections of the specimen were measured using 10 linear variable differential transducers (LVDTs) at mid-span, quarter-span, and side spans. Two transducers were symmetrically mounted in the transverse direction to check load eccentricity. Strain gauges were arranged at mid span and quarter spans as presented in Figure 6b,c. As the concrete was casted into the steel girder, the slip in the interface between concrete and steel could not be monitored directly using the steel–concrete interlayer slip sensor. For this reason, in this paper, the interlayer slip was revealed indirectly by the strain gauges deployed in the specimen.

## 3. Experimental Results

### 3.1. Load–Deflection Relationship and Failure Mode

The load–deflection curves at mid-span and quarter- span are shown in Figure 7. The deflection value is the average of the two transversally symmetric LVDTs. It can be inferred from Figure 7 that the deflection at the quarter spans were symmetric with each other. The curves generally include an elastic stage, a plastic hardening stage and a final unloading phase. The initial bending stiffness (at 100 kN) of the specimen at mid span and quarter span was 61.1 kN/mm and 83.0 kN/mm, respectively. The sound of the interlayer slip between the steel and concrete appeared at 140 kN. As load kept increasing, steel yielding happened at mid-span and led to nonlinear deflection development. The steel top flange began to buckle at the load level of 780 kN, while the specimen deflection grew drastically. The specimen completely lost carrying capacity at 975.4 kN due to the fracture of the steel bottom flange.

The main cause of the specimen’s destruction was the rupture of the steel bottom flange along with steel local buckling in the upper flange, as presented in Figure 8. The specimen was cut through the mid-span after loading to observe the failure mode of the concrete girder. As presented in Figure 9, the steel top and bottom flange was detached from the concrete, and the concrete web was crushed. In addition, diagonal cracks appeared in the concrete web and flanges. In general, the concrete girder prevented the steel top flange from extensive buckling, which significantly improved the load-carrying capacity of the specimen. In addition, strengthening the interaction between the steel and concrete helped to prevent the interlayer slip, which simultaneously contributed to a higher flexural stiffness.

### 3.2. Strains in Steel Flange and Reinforcing Bars

Figure 10 shows the load–strain curves of the steel flanges at quarter-span and mid-span. It can be inferred that both steel top and bottom flanges reached yielding at mid-span, while only steel bottom flange yielded at the quarter-span. In addition, there was an obvious phenomenon of steel–concrete interlayer slip both at mid-span (at 140 kN) and quarter-spans (at 240 kN) in the steel bottom flange, confirming what was visually observed after the test.

The load–strain curves of the steel web and the longitudinal reinforcing bars at mid-span are shown in Figure 11 (some strain gauges on reinforcing bars were damaged during construction and are not shown). Strain gauges N2-W-1, N2-R2-1/2, N2-W-2 and N2-R3-1/2 share the same height as the section. Regarding the strain gauge N2-W-1, N2-R2-1/2, N2-W-2 and N2-R3-2, it is evident that the strain of the steel web and reinforcing bars at the same section height is identical during the elastic stage; however, at 200 kN, there is an interlayer slip observed in the nethermost reinforcing bar. The last two rows of reinforcing bars reached yielding at the later loading stage, while the rest remained elastic.

### 3.3. Sectional Strain Distribution

The sectional strain distribution under different load stages at mid-span is shown in Figure 12. The sectional strain distribution up to 100 kN matched well with the plane section assumption from the elastic beam theory. The strain at the bottom of the girder had an uprush at the load stage of 200 kN due to the steel–concrete interlayer slip. The phenomenon of interlayer slip matched with the test observation and the strain monitoring. With the increase of the load, the interlayer slip gradually stretched upward.

As shown in Figure 6c, the steel top flange was divided into four cells by longitudinal stiffeners, and in the mid-span section, strain gauges were deployed both in the support and in the center part of these cells. Figure 13 shows the strain versus loading at the cell supports and cell central part. It can be inferred that the strain profile across the top steel flange is uniform up to 400 kN. As the load kept increasing, the strain at the different locations tends to diverge due to minor local buckling. Generally, the stiffening rib determined the position of the local buckling but did not have a significant impact on the strain variation during the loading.

## 4. Numerical Analysis

### 4.1. Numerical Simulation of Composite Girder

A finite element model was developed using Abaqus [29] to further study the structural performance of the composite steel–concrete joint of the hybrid beam bridge. Figure 14 shows the element types and meshes of each component in the model. The concrete girder, steel girder and reinforcing bars were discretized by solid element C3D8R, shell element S4R and truss element T3D2, respectively. The contact between steel and concrete was simulated by penalty frictional formulation along a tangential direction and ‘Hard’ contact along a normal direction. The friction coefficient was set as 0.1 [30]. The reinforcing bars were constrained with the concrete girder and the stiffening rib in the corresponding position. The loading scheme adopted “uniform” loading in Abaqus, which evenly distributed the load on the loading surface. The displacement boundary conditions took linear constraint support, where all the elements’ nodes along the constraint line were restricted. The model comprised a total of 48,050 elements and 56,355 nodes. Dynamic explicit analysis method was employed in the analysis process, while the calculation can be completed within 50 min using a 64-core workstation.

In order to ensure the model’s accuracy, sensitivity analyses were conducted on the finite element modelization and presented in Table 1. All eight models listed in the table demonstrated good convergence, as indicated by the values of $A_{e}/I_{e}$ and $K_{e}/I_{e}$ being lower than 5%. The results showed that the dilation angle, layers of concrete and constitutive model of concrete had minimal impact on the accuracy of the model. To simplify the meshing work and divide the concrete slab into four layers, a mesh size of 50 mm was used for both concrete and steel. Based on the results, the parameters of the second model in Table 1 were determined for the simulation.

### 4.2. Material Properties

The concrete damage plasticity model [31,32,33] was used for concrete. Figure 15 presents the uniaxial stress–strain curve as determined according to GB50010-2010 [34] using Equation (1) to Equation (6). Equations (2) and (5) describe the coefficients of $d_{t}$ and $d_{c}$. Parameters $\alpha_{t}$ and $\alpha_{c}$ were the regulation coefficients of the descending part, which are both empirical parameters as provided in GB50010 [34]. The plastic strain input into Abaqus equals the total strain minus the elastic strain, where $\varepsilon_{t,r}$ and 0.4$f_{c,r}$ were taken as the tensile and compressive elastic strains, respectively [35]. Equation (7) defines the damage factor *d* (input into Abaqus) proposed by Sidoroff [36].
(1)$$\sigma =(1-d_{t})E_{c}\varepsilon ,$$
(2)$$d_{t}=\left\{\begin{array}{cc} 1-\rho_{t}[1.2-0.2x^{5}] & x\leq 1\\ 1-\frac{\rho_{t}}{\alpha_{t}{\left(x-1\right)}^{1.7}+x} & x>1 \end{array}\right.,$$
(3)$$\mathrm{where}\;x=\frac{\varepsilon }{\varepsilon_{t,r}},\;\rho_{t}=\frac{f_{t,r}}{E_{c}\varepsilon_{t,r}}.$$
(4)$$\sigma =(1-d_{c})E_{c}\varepsilon ,$$
(5)$$d_{c}=\left\{\begin{array}{cc} 1-\frac{\rho_{c}n}{n-1+x^{n}} & x\leq 1\\ 1-\frac{\rho_{c}}{\alpha_{c}{\left(x-1\right)}^{2}+x} & x>1 \end{array}\right.,$$
(6)$$\mathrm{where}\;x=\frac{\varepsilon }{\varepsilon_{c,r}},\;\rho_{c}=\frac{f_{c,r}}{E_{c}\varepsilon_{c,r}},\;n=\frac{E_{c}\varepsilon_{c,r}}{E_{c}\varepsilon_{c,r}-f_{c,r}}.$$
(7)$$d=1-\sqrt{\frac{\sigma }{E_{0}\varepsilon }}$$

*E_c_*: Elasticity modulus of concrete.

$\alpha_{t}$ and $\alpha_{c}$: Regulation coefficients of the descending part of concrete.

*d_c_* and *d_t_*: Compressive and tensile damage factor of the concrete.

*f_t,r_* and *f_c,r_*: Ultimate tensile and compressive strengths of concrete, respectively.

$\varepsilon_{t,r}$ and $\varepsilon_{c,r}$: Strains corresponding to the ultimate tensile and compressive strengths of concrete, respectively.

Figure 16a presents a trilinear stress–strain curve for steel plates. The yielding and ultimate stresses $\sigma_{ys}$ and $\sigma_{us}$ were obtained from the material property tests. The ultimate strain $\varepsilon_{us}$ was 0.6%, while the yield strain $\varepsilon_{ys}$ was derived using $\sigma_{ys}/E_{s}$. A bilinear stress–strain model was adopted for the reinforcing bar, as shown in Figure 16b. As the reinforcing bar exhibited no evident yielding point, the yield stress took the ultimate strength from the material property test.

### 4.3. Validation of FEM Analysis

The load–deflection curve from the FEM model is compared with the test result in Figure 17. The two curves matched well with each other. The phenomenon of interlayer slip around 150 kN appeared in the theoretical curve, which is consistent with the experimental observation and the strain analysis. The initial bending stiffness (at 100 kN) and the load-carrying capacity of the FEM model were 72.4 kN/mm and 1009.6 kN, respectively, which were slightly larger compared to the test specimen. The deviation could be attributed to the accuracy of the material constitutive laws as well as to the simulation of the steel–concrete interlayer interaction.

### 4.4. Influence of Composite Action on Flexural Behaviour

Models with solo concrete sections and solo steel sections were established to study the influence of composite action on the flexural behavior. Parametric analysis varying the thickness of the steel plates of the specimen with a solo steel section was also conducted. The rest of parameters in the numerical models remained unchanged from the test specimen.

Figure 18a shows the influence of the composite action on the load-carrying capacity. The load-carrying capacity of the specimen with composite action is much larger than that of specimens with solo concrete or solo steel, and even much larger than the sum of both (see OC + OS3 value in the figure). The huge difference is due to the failure mode of the steel girder. As shown in Figure 18c, the failure mode of the specimen with a solo steel section was local buckling, which did not appear in the specimen with a composite section. As the steel girder provided greater load-carrying capacity, the concrete in the composite section prevented local buckling, which significantly increased the load-carrying capacity of the joint section. It can also be deduced from Figure 18a that adopting a composite section saves around 67% steel flange thickness compared to adopting a solo steel section.

The influence of composite action on initial stiffness (during loading up to 100 kN) is shown in Figure 18b. The increase of steel flange thickness in the specimen with solo steel section had limited influence on the initial stiffness, while the composite section could increase the initial stiffness by nearly three times (compared with OS3).

OC: Specimen with a concrete section only.OSN: Specimen with a steel section only and a steel plate thickness equal to N mm.COM: Specimen with a composite section and a steel plate thickness equal to 3 mm.

### 4.5. Parametric Analysis on Steel Flange Thickness

Models with different steel flange thickness (the other section parameter kept the same as in the test specimen) were established to study its influence on the flexural behavior. Figure 19a shows that the load-carrying capacity presented a linear relation with the steel flange thickness, indicating that the flexural strength depended mainly on the steel girder. As the main failure mode of the test specimen was steel flange fracture, this conclusion is consistent with the experimental results. In other words, the primary function of the concrete filling was to prevent local buckling of the steel, while it had limited influence on the load-carrying capacity of the joint section. In addition, the increase of the steel flange thickness also contributed to an improvement of the initial stiffness (as shown in Figure 19b).

## 5. Conclusions

A simply supported composite girder modelling the full section of a prototype hybrid bridge was fabricated and tested in this study. Finite element models of the specimen were built and calibrated for the posterior simulation and parametric analysis. Based on the experimental and numerical results, the following conclusions were drawn:According to the test results, the failure mode of the specimen was fracture of the steel bottom flange along with steel local buckling of the upper flange. The stiffening ribs on the upper flange determined the position of the local buckling but did not have a significant impact on the strain variation during the loading. The concrete girder prevented the steel flange from extensive buckling, which significantly improved the load-carrying capacity of the specimen.According to the test, there was an obvious phenomenon of steel–concrete interlayer slip both at mid-span (at 140 kN) and quarter-span (at 240 kN) in the steel bottom flange. The steel bottom flange was observed to be completely detached from the concrete. Thus, strengthening the interaction between the steel and concrete helps to prevent the interlayer slip and simultaneously contributes to a higher flexural stiffness.Based on the numerical analysis, the FEM model was proved to be reliable with sufficient accuracy. The deviation from the experimental results could be attributed to the lack of accuracy of the material constitutive laws as well as to the not fully correct simulation of the steel–concrete interlayer interaction.Based on the numerical parametric analysis, it is observed how the load-carrying capacity of the specimen with a composite section is much larger than that of specimens with a solo concrete sections and solo steel sections. While the steel girder provided greater load-carrying capacity, the concrete in the composite section prevented local buckling and increased the load-carrying capacity of the joint section. In addition, an increase of steel flange thickness in the specimen with a solo steel section had limited influence on the initial stiffness, while the composite section drastically increases it.Based on the parametric analysis, the load-carrying capacity showed a linear relation with the steel flange thickness, indicating that the flexural strength depended mainly on the steel girder. As the main failure mode of the test specimen was steel flange fracture, this conclusion is consistent with the experimental results. Thus, the primary function of the concrete filling was to prevent the steel local buckling, while it had limited influence on the load-carrying capacity of the joint section. In addition, the increase of the steel flange thickness also contributed to an improvement of the initial stiffness.

Although some relevant conclusions have been already obtained from the study shown in this paper, still some limitations of the presented research have to be pointed out. For instance, for financial reasons, only one specimen was tested in this study. More test specimens with different design parameters are considered in the subsequent studies to obtain additional test data. In addition, as the test was designed to model the global performance and strength of the proposed connection, the small-scale model is hard to analyze the phenomenon of local buckling on the top flange that is due to local stress of the composite section of the specimen. Other local failure mechanisms, for instance the detachment of the steel and concrete in the top and bottom flanges, are due to the constraints imposed by the design of a small-scale specimen. Whether these local mechanisms can occur in the hybrid girder should be investigated by a larger scale test in the future study.

## Figures and Tables

**Figure 1 materials-16-03265-f001:**
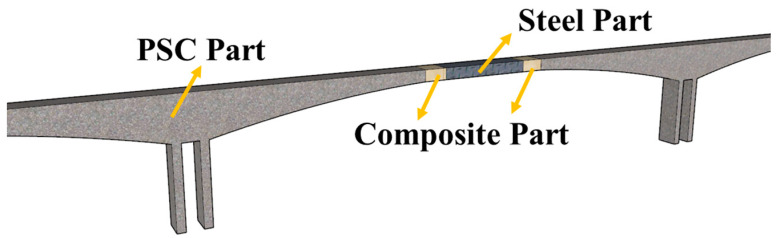
Illustration of a hybrid continuous bridge.

**Figure 2 materials-16-03265-f002:**
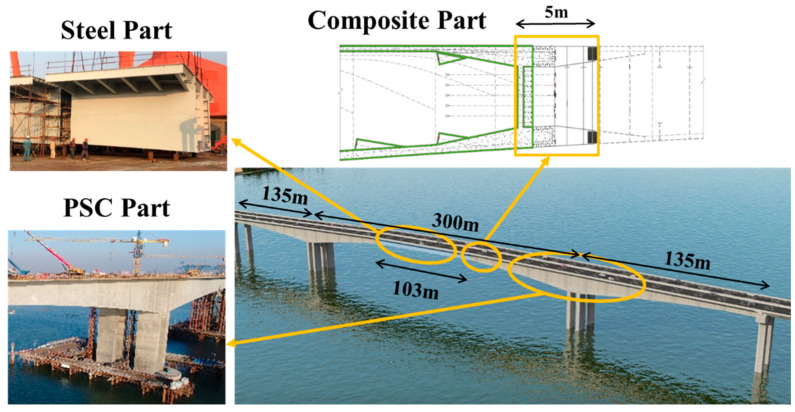
Configuration of Anhai Bay Bridge.

**Figure 3 materials-16-03265-f003:**
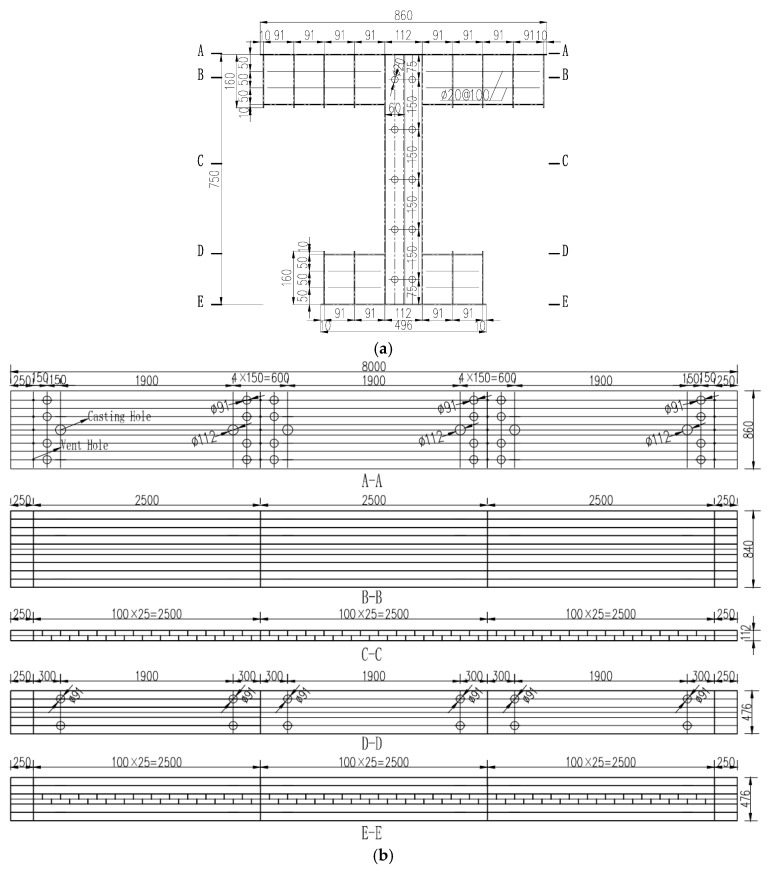
Configuration of the test specimen (mm). (**a**) Composite section and reinforcing bar layout (mm); (**b**) specimen definition.

**Figure 4 materials-16-03265-f004:**
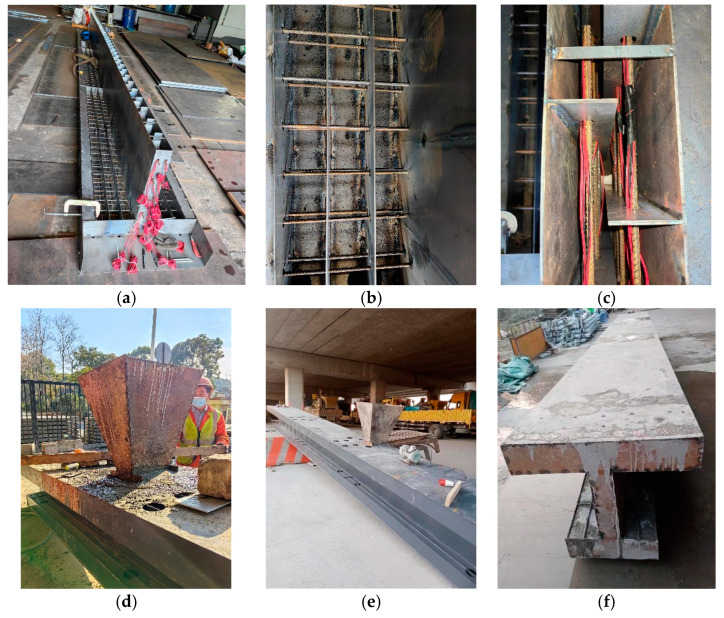
Photos during specimen construction. (**a**) Steel girder; (**b**) transversal reinforcing bars; (**c**) longitudinal reinforcing bars; (**d**) casting concrete; (**e**) vibrating concrete; and (**f**) fabricated specimen.

**Figure 5 materials-16-03265-f005:**
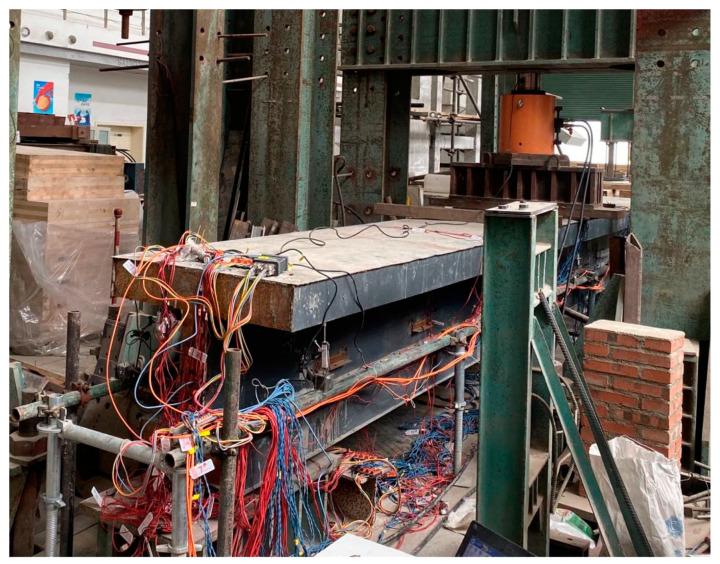
Loading setup.

**Figure 6 materials-16-03265-f006:**
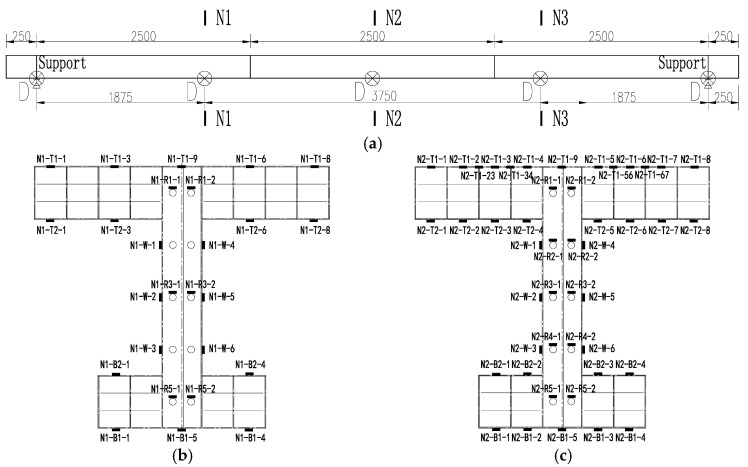
Sensor positions (mm). (**a**) Identification of sections of interest (mm), D: linear variable differential transducers (LVDT); (**b**) strain gauge arrangement at quarter-span; (**c**) strain gauge arrangement at mid-span.

**Figure 7 materials-16-03265-f007:**
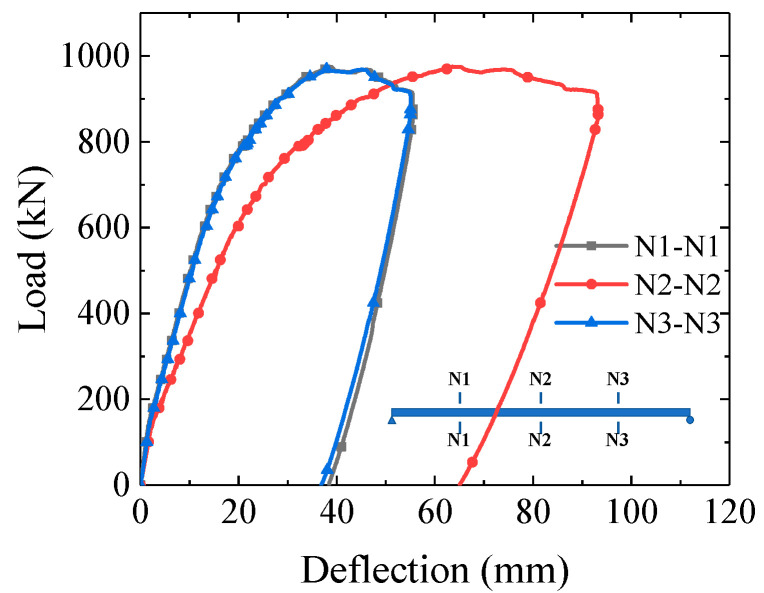
Load–deflection curves at mid-span and quarter-span.

**Figure 8 materials-16-03265-f008:**
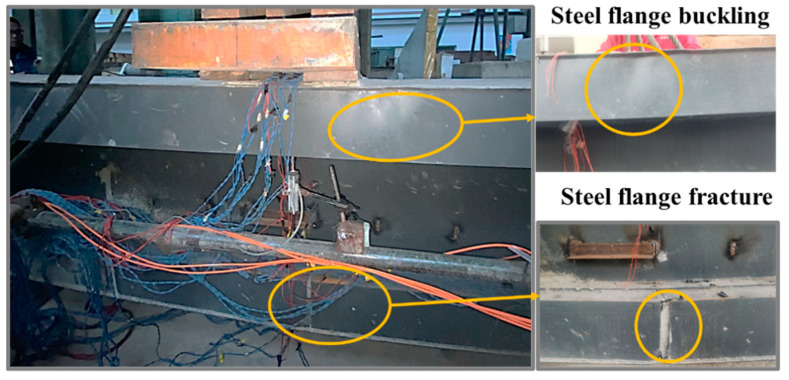
Failure mode of the steel part of the girder.

**Figure 9 materials-16-03265-f009:**
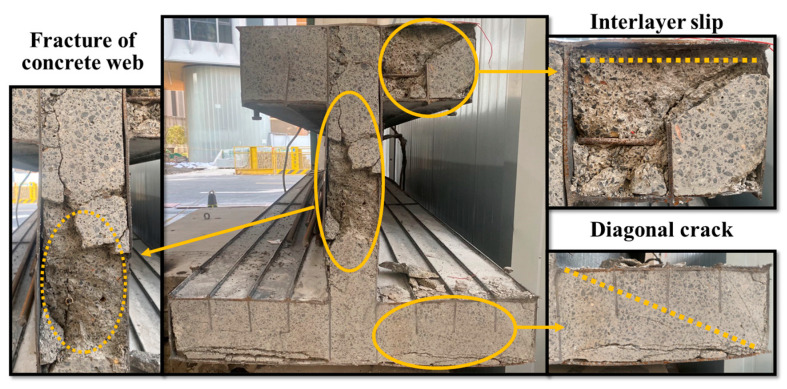
Failure mode of the concrete part of the girder.

**Figure 10 materials-16-03265-f010:**
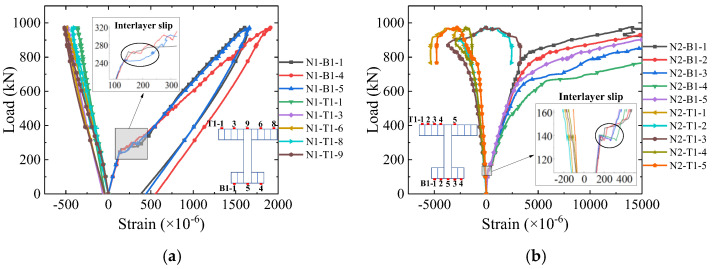
Load–strain curves of the steel flange. (**a**) Quarter-span; (**b**) mid-span.

**Figure 11 materials-16-03265-f011:**
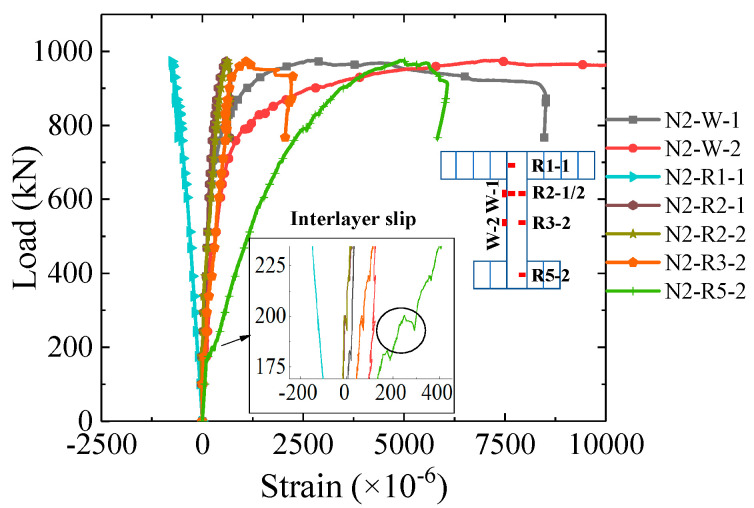
Load–strain curves of the reinforcing bars at mid-span.

**Figure 12 materials-16-03265-f012:**
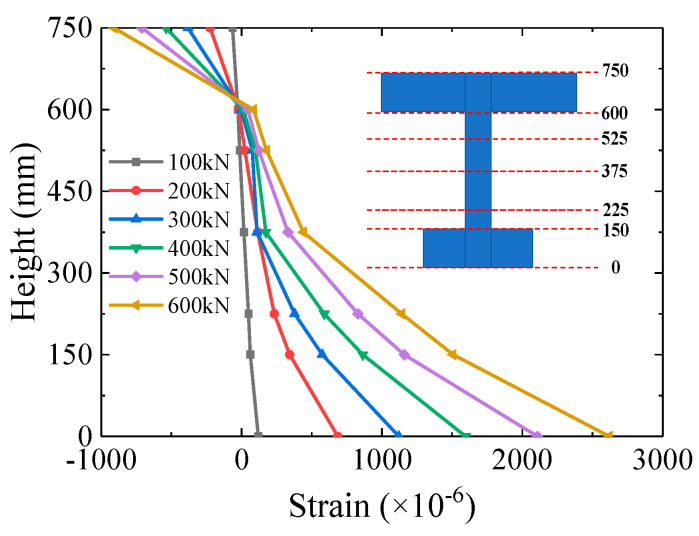
Distribution of sectional strain at mid-span.

**Figure 13 materials-16-03265-f013:**
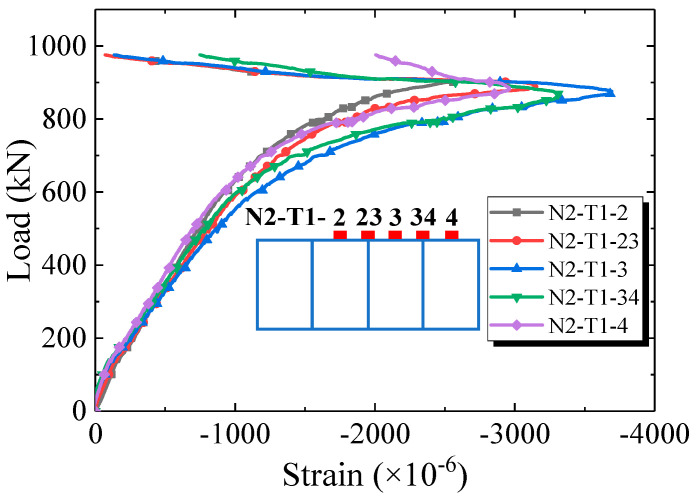
Load–strain curves at support and center of the cells at mid-span.

**Figure 14 materials-16-03265-f014:**
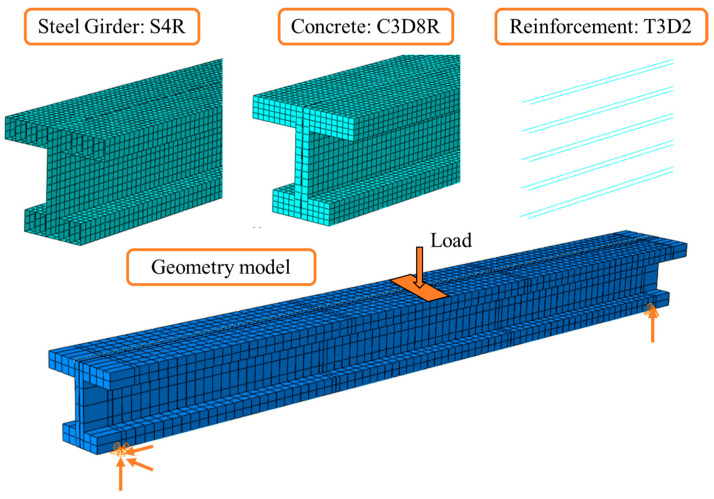
Discretization, element types and loading and boundary conditions in the FEM model.

**Figure 15 materials-16-03265-f015:**
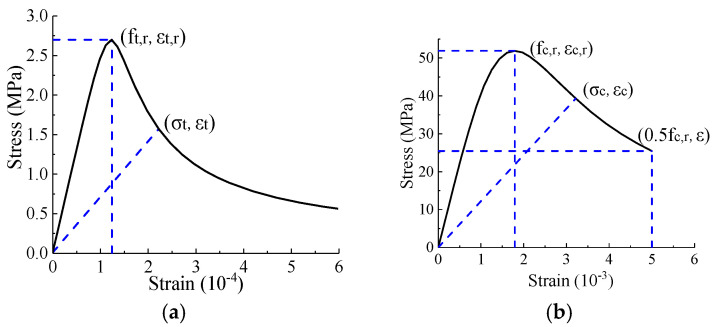
Stress–strain curve for concrete. (**a**) Tensile stress; (**b**) compressive stress.

**Figure 16 materials-16-03265-f016:**
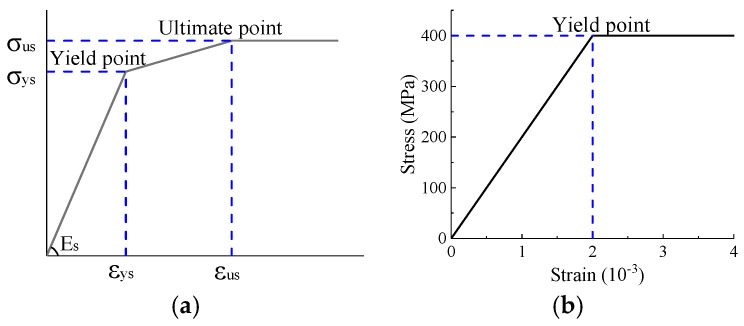
Stress–strain model for steels. (**a**) Steel plate; (**b**) reinforcing bar.

**Figure 17 materials-16-03265-f017:**
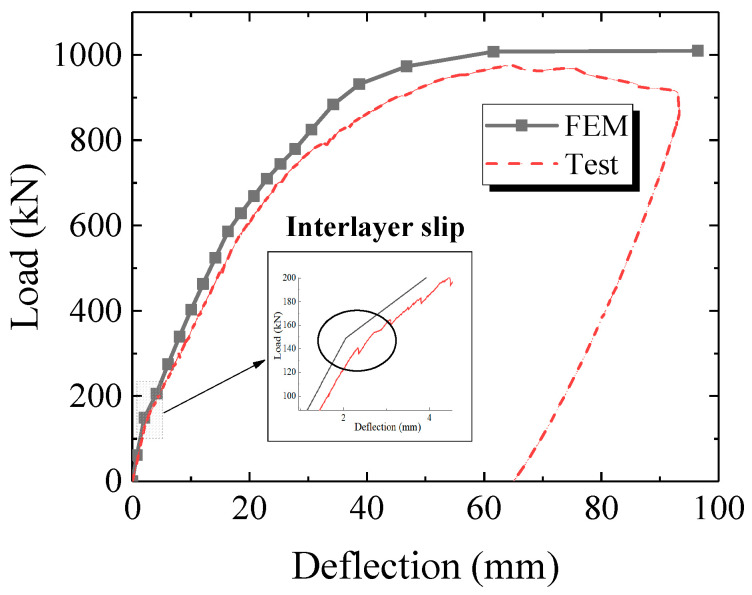
Comparison of numerical and experimental load–deflection curves at mid-span.

**Figure 18 materials-16-03265-f018:**
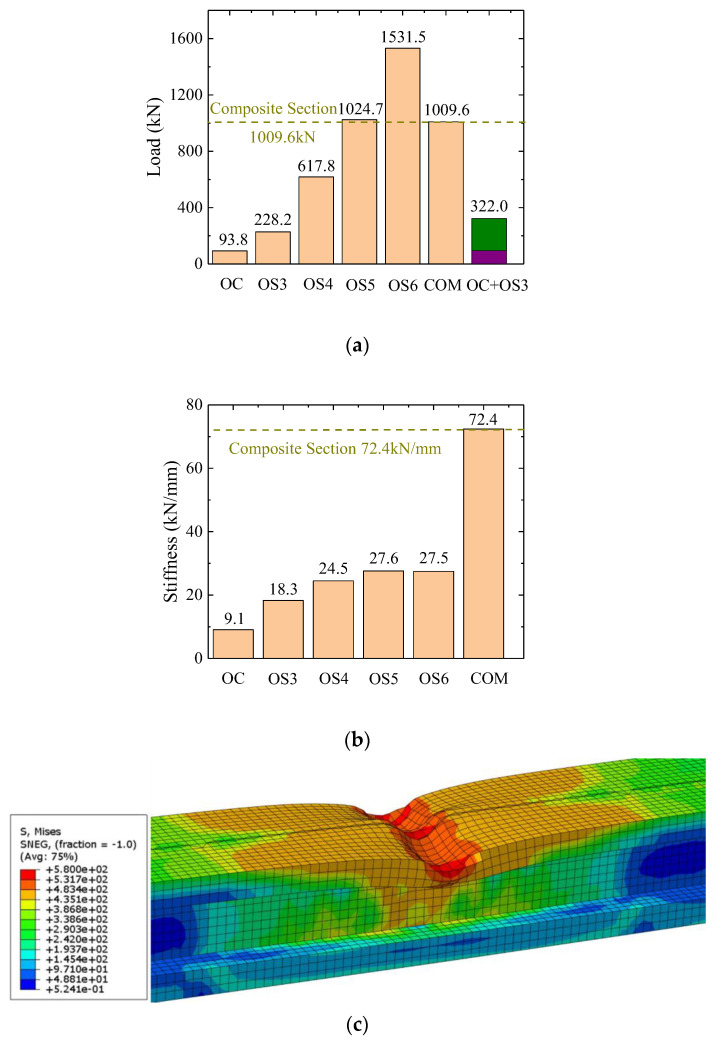
Influence of composite action on flexural behavior. (**a**) Influence of composite action on load-carrying capacity; (**b**) influence of composite action on initial stiffness; (**c**) failure mode of specimen OS6.

**Figure 19 materials-16-03265-f019:**
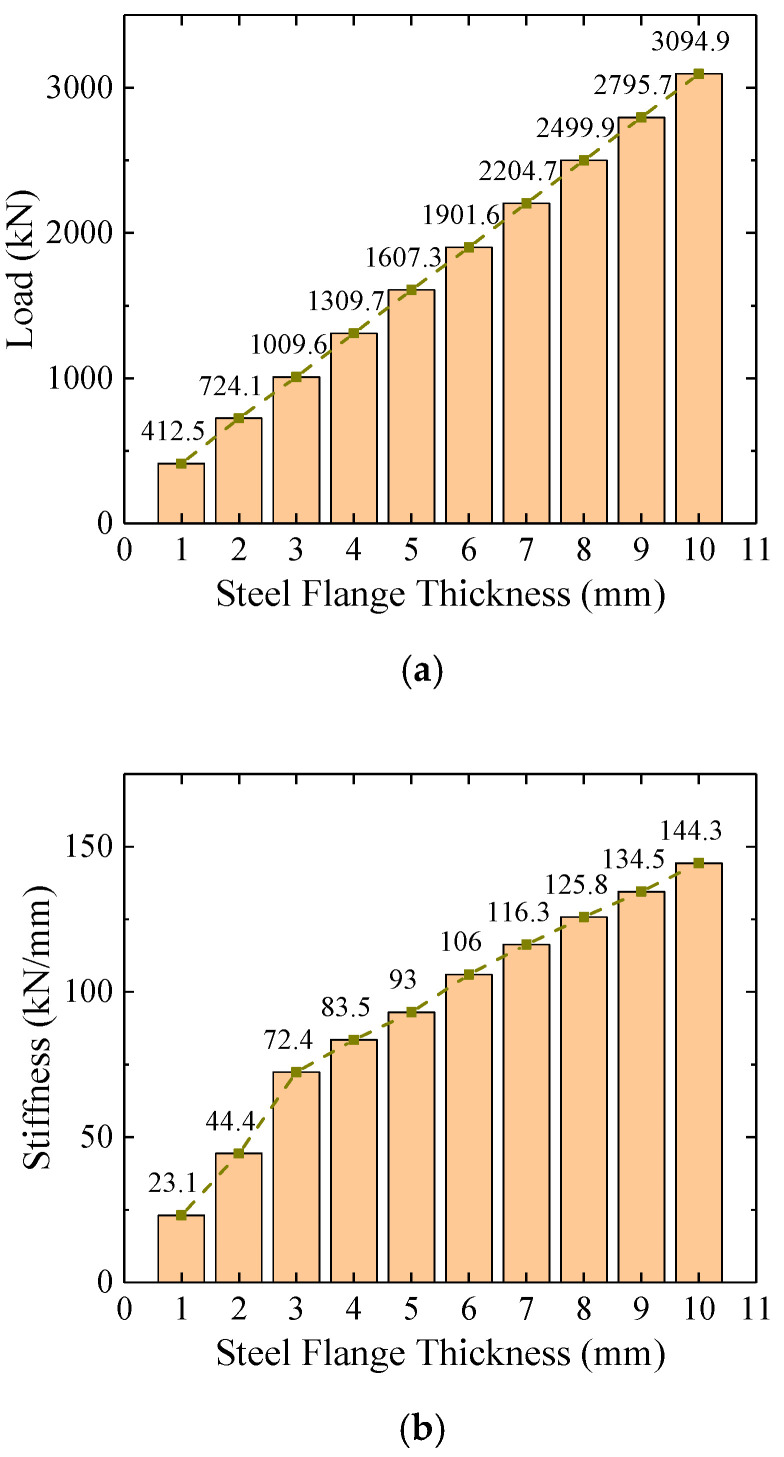
Influence of steel flange thickness on flexural behavior. (**a**) Influence of steel flange thickness on load-carrying capacity; (**b**) influence of steel flange thickness on initial stiffness.

**Table 1 materials-16-03265-t001:** Results of sensitivity analysis.

No.	Dilation Angle (°)	Concrete Layers	Constitutive Model	Mesh Size (mm)		$A_{e}/I_{e}$ **(%)**	$K_{e}/I_{e}$ **(%)**
				Concrete	Steel		
1	25	4	fib	50	50	4.1%	2.7%
2	37.5	4	fib	50	50	1.9%	1.3%
3	50	4	fib	50	50	1.5%	1.4%
4	37.5	4	fib	50	50	3.3%	1.6%
5	37.5	4	fib	50	50	3.9%	1.9%
6	37.5	2	fib	100	100	1.6%	0.8%
7	37.5	4	fib	50	100	1.5%	1.0%
8	37.5	4	Secant Modulus	50	50	3.9%	1.9%

## Data Availability

Some or all data that support the findings of this study are available from the corresponding author upon reasonable request.

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
