# Peer review of "Modeling and Testing of a Composite Steel–Concrete Joint for Hybrid Girder Bridges"

_materials, 2023, doi:10.3390/ma16083265_

Round 1
Reviewer 1 Report
A good attempt has been made in this paper to develop a new work, I recommend its acceptance, as a minor suggestion,
Limitation of the work should be clearly stated in the text, for example limited number of case studies and limitation of proposed path in practice.
Author Response
Thanks a lot for the reviewer’s instructive advices. For financial reasons, only one specimen was fabricated and loaded in this study. More test specimens with different design parameters are considered in the subsequent studies to obtain more test data. In addition, the small-scale model is hard to analyze the local stress of the composite section in the hybrid girder, which can be supplemented by a larger scale test in the future study. We have explained this limitation at the end of the conclusion part.
Reviewer 2 Report
The topic of the paper is very interesting from practical point of view. This is something that is frequently faced in the design of long span bridges. However, there are still some comments that need consideration to improve the paper. Here is the list of comments:
1. References 1-4 (line 32) does not really say about the web cracking. Please clarify.
2. The test model is said to be 1:8 scale. Since this is a small-scale model, please describe how you achieve the similitude relationship between the test model and the prototype (both for the geometry and the material and also for the load). Please also put emphasize on the similitude requirement in the interface.
3. The specimen used for the study was only one test specimen? Why?
4. Please justify the similitude of using different cross section in the test model (from that used in the prototype bridge)
5. The composite portion is not clearly described. For example, how it is connected in the interface? By shear connectors? How is the transfer mechanisms in that interface? How it is calculated?
6. (Line 122) --> Is there any concrete cube specimen tested at the same time as the test of the model test specimen?
7. (Line 128) --> the specimen seems to be tested with three point bending test procedure. Why? By doing so, you don't have a region with a constant moment and most of the test region is becoming disturbed. How you then make sure that the failure will occur right at the midspan?
8. Please describe hoow you monitored the slip in the interface between concrete and steel?
9. Why the top flange buckled? Is it expected in the design of the test specimen? Why you allow buckling as the limit state in this case? (Line 154)
10. Why the top and bottom flanges are also detached from concrete? Is it allowed in the design to have this mechanism as the limit state?
11. Please elaborate line 185-186
12. The sectional strain in the midspan is not linearly distributed, even in the elastic range. Why? (Figure 12)
13. Line 219-222 --> the load transfer mechanism in the interface zone is only through friction mechanism? How the rebars crossing the interface contribute to this mechanism?
14. Figure 17 --> Curve from the test results shows many points of degradation during the loading. While the analytical curve does not show that (it is smoother). Please describe why?
15. Figure 18 --> is it acceptable mode of failure? Please explain...
Other comments:
1. Line 54, the name of the author is different between the one mentioned in the text and the one in the Reference list (No. 17)
Author Response
Thanks a lot for the reviewer’s instructive advices. All the comments and suggestions have been carefully considered and revised in the resubmitted manuscript. Meanwhile, the point-to-point responses to the reviewers' comments are listed as follows.
- References 1-4 (line 32) does not really say about the web cracking. Please clarify.
Response: References 1-4 presents the general mechanical behavior and common disadvantages of prestressed concrete girder bridges. We have changed the position of the references to make it clearer. Please refer to Section 1 in the revised manuscript.
- The test model is said to be 1:8 scale. Since this is a small-scale model, please describe how you achieve the similitude relationship between the test model and the prototype (both for the geometry and the material and also for the load). Please also put emphasize on the similitude requirement in the interface.
Response: The girder test aimed to obtain the sectional bending capacity of the specimen. Thus, all the section geometric parameters of the specimen were determined as one-eighth of the prototype bridge. (For example, as the girder height, steel plate thickness and web width of the prototype bridge were respectively 6000mm, 24mm and 900mm, then those of the specimen were determined as 750mm, 3mm, 112mm.) We have added the illustration to make it clearer. Please refer to Section 2.1 in the revised manuscript.
The specimen material was the same as the prototype bridge. Considering that the yield stress of the prototype bridge and the test specimen are the same, the sectional moment carrying capacity of the prototype bridge is approximately 512 times of the test specimen. Common arrangement of stud connectors can achieve a full connection between the steel-concrete interlayer [1]. Thus, the interlayer in the specimen was also as to be fully connected. As the specimen was too small to install shear connectors, structural form of perfobond rib was adopted to create the steel-concrete full connection.
[1] H. Su, Q. Su, C. Xu, X. Zhang, and D. Lei, "Shear performance and dimension rationalization study on the rubber sleeved stud connector in continuous composite girder," Engineering Structures, Article vol. 240, Aug 1 2021, Art. no. 112371.
- The specimen used for the study was only one test specimen? Why?
Response: For financial reasons, only one specimen was tested in this study. More test specimens with different design parameters are considered in the subsequent studies to obtain more test data. In addition, in the small-scale model is hard to analyze the local stress of the composite section in the hybrid girder, which can be supplemented by a larger scale test in the future study. We have explained this limitation at the end of the conclusion part.
- Please justify the similitude of using different cross section in the test model (from that used in the prototype bridge)
Response: The cross section of the prototype bridge is a box section. However, as the test model is a small-scale model, the two webs of the box section will be too thin to fabricate. Thus, the specimen combined the two webs together. As is illustrated in the response to question 2, all the sectional parameters of the specimen were determined as one-eighth of the prototype bridge. Thus, the sectional bending capacity of the specimen can be converted to that of the prototype bridge. We have added the explanation to make it clearer. Please refer to Section 2.1 in the revised manuscript.
- The composite portion is not clearly described. For example, how it is connected in the interface? By shear connectors? How is the transfer mechanisms in that interface? How it is calculated?
Response: The specimen was too small to install shear connectors. After welding shear connectors, the concrete would be harder to cast into the steel girder. Besides, welding shear connectors on steel plates with thickness of only 3mm will generate obvious residual stress. Thus, structural form of perfobond rib was adopted for the steel-concrete connection (as is presented in Figure 4 (b) and Figure 4 (c)). Besides, the dense transverse ribs also helped to increase the friction so as to increase the connection in the interface. We have added an explanation to make it clearer. Please refer to Section 2.1 in the revised manuscript.
- (Line 122) --> Is there any concrete cube specimen tested at the same time as the test of the model test specimen?
Response: We only tested the properties of the concrete on the 28th day after casting, according to the Chinese Standard for test method of mechanical properties on ordinary concrete (GB/T 50081-2002). Truly it will be more accurate to conduct a material test at the same time as the test of the specimen. However, we think that the concrete compressive strength between the day of the material test and the specimen test will be very similar as there are only 15 days of difference. Therefore, the error is negligible. We have added the reference to the standard. Please refer to Section 2.2 and list of references in the revised manuscript.
- (Line 128) --> the specimen seems to be tested with three point bending test procedure. Why? By doing so, you don't have a region with a constant moment and most of the test region is becoming disturbed. How you then make sure that the failure will occur right at the midspan?
Response: We agree with the reviewer that a four-point bending test provides a constant moment region, however this does not guarantee that the failure will be exactly at the mid-span, due to the un-homogeneity of the concrete material and variability of mechanical properties of steel and concrete, apart from the geometrical construction tolerances. During the test, the deflection and strain along the transverse and longitudinal directions were continuously monitored. As is presented in Figure 7, the deflection at quarter spans were symmetric with each other, what more or less guarantees that the applied load is in a symmetric location (in the mid-span section). Because of this, in fact, we achieved the failure in the mid-span section.
- Please describe how you monitored the slip in the interface between concrete and steel?
Response: As the concrete was casted into the steel girder, the slip in the interface between concrete and steel could not be monitored directly using the steel–concrete interlayer slip sensor. For this reason, the interlayer slip was revealed indirectly by the strain gauges deployed in the specimen. We have added this explanation in the revised manuscript.
- Why the top flange buckled? Is it expected in the design of the test specimen? Why you allow buckling as the limit state in this case? (Line 154)
Response: The buckling was due to the separation between the steel and concrete, and was not expected in the design of the test specimen. Therefore, buckling was not the limit state considered in this case. Indeed, adopting shear connectors might prevent the buckling in the top flange. However, the top flange buckled at the end of loading. The phenomenon of buckling was subtle and had little influence on the load-carrying capacity of the test specimen. That is because the main failure mode was the steel bottom flange fracture, when the top flange buckling was still not fully developed. Generally, in the small-scale model, it is hard to analyze this local phenomenon due to the local stress of the composite section in the hybrid girder, as we have explained at the end of the conclusions in the revised manuscript.
- Why the top and bottom flanges are also detached from concrete? Is it allowed in the design to have this mechanism as the limit state?
Response: Again, in the design this detachment mechanism was not considered as a limit state. As is illustrated in question 5, the specimen was too small to install shear connectors. Structural form of perfobond rib was adopted for the steel-concrete connection (as is presented in Figure 4 (b) and Figure 4 (c)). Thus, there was a reliable connection internally between the steel and concrete, but the connection at the surface was relatively weaker. The research presented in this paper aimed to reveal the overall mechanical performance of the composite section in the hybrid girder, but the local mechanical behavior is hard to analyze with the small-scale model. The local mechanical behaviour leading to the detachment of concrete and steel in the flanges can be supplemented and investigated by a larger scale test in the future study. This has been added in the conclusions
- Please elaborate line 185-186
Response: We have elaborated this part. Please refer to Section 3.1 in the revised manuscript.
- The sectional strain in the midspan is not linearly distributed, even in the elastic range. Why? (Figure 12)
Response: As presented in Figure 12, the sectional strain in the midspan is linearly distributed before load stage of 100 kN. However, the strain at the bottom of the girder had an uprush after the load stage of 200 kN due to the steel-concrete interlayer slip. The phenomenon of interlayer slip matched with the test observation and the strain monitoring. We have added the explanation. Please refer to Section 3.3 in the revised manuscript.
- Line 219-222 --> the load transfer mechanism in the interface zone is only through friction mechanism? How the rebars crossing the interface contribute to this mechanism?
Response:
The contact between steel and concrete were simulated by penalty frictional formulation along tangential direction and ‘Hard’ contact along normal direction. Thus, besides the friction mechanism, all sides of the concrete filling were restrained by steel plates. The transverse and longitudinal reinforcing bars were constrained with the concrete girder and the stiffening rib in the corresponding position, but didn’t contribute to the friction mechanism. We have added the explanations, please refer to Section 4.1 in the revised manuscript.
- Figure 17 --> Curve from the test results shows many points of degradation during the loading. While the analytical curve does not show that (it is smoother). Please describe why?
Response: The points of degradation were due to the stop intervals during the loading process. While loading the test specimen, there was an interval stop every 20kN for taking photos. During the stop interval, the load showed a temporary downward trend. However, there wasn’t such intervals in the numerical analysis. Thus, the analytical curve is smoother.
- Figure 18 --> is it acceptable mode of failure? Please explain.
Response: Figure 18 aimed to illustrate the influence of the concrete filling on the load-carrying capacity of the specimen, and the local buckling is possible to occur in the specimens with solo steel section when the steel plate thickness is too small.
Figure 18 (c) presents the failure mode of specimen OS6, which had steel section only (the steel plate thickness was 6mm) and was not filled with concrete. Local buckling occurred around the loading position in specimen OS6, which significantly decreased the specimen’s load-carrying capacity. In comparison, the local buckling did not appear in the specimen with composite section. Thus, the concrete in the composite section contributed to a prevention of the local buckling, which significantly increased the load-carrying capacity of the joint section.
- Line 54, the name of the author is different between the one mentioned in the text and the one in the Reference list (No. 17)
Response: Thanks to the reviewer for noticing this error. We have revised the author’s name in the text. Please refer to Section 1 in the revised manuscript.
Reviewer 3 Report
This paper is well-written and organised. The research methodology adopted and described is robust. The proposed study represents an advance concerning the state-of-the-art on this topic.
Before publication, some specific comments are given to the authors to enhance and clarify some parts of their manuscript.
1. Introduction
The introduction should be proposed according to a more holistic approach. To this scope, the authors should mention that the analysis of steel-to-concrete connections in hybrid beams (stress transfer mechanisms, seismic response, etc.) refers not only to the beams and applications mentioned in their paper but also to classical composite beams, box beams and non-standard composite beams such as hybrid trussed beams. The latter, usually adopted for civil structures and infrastructures (e.g., https://doi.org/10.1016/j.engstruct.2012.11.004), in the form of full- and small-thick geometry (e.g., https://doi.org/10.1016/j.conbuildmat.2017.11.134) with different failure modes. These girders have been studied in recent years and analyzed by several authors through detailed finite element contact models and theoretical formulations that evidence how different steel-concrete interactions can determine different failure modes in the beam. Also, the size effect on the structural performance should be mentioned, which is typically evaluated through numerical models when laboratory tests cannot be performed. Finally, the seismic damping capacity should be treated (e.g., http://ingegneriasismica.org/dissipative-connections-of-rc-frames-with-prefabricated-steel-trussed-concrete-beams)
2.1 Specimen design
With reference to Figures 3a) and b), please improve the readability of the quotes of holes.
2.3 Loading and monitoring setup
With reference to Figure 6a), maybe it would be better to improve the indicators of the Strain gauge arranged at quarter-span because it is not actually placed at the quarter-span in your drawing.
3.2 Strains in steel flange and reinforcing bars
With reference to Figure 10 and Figure 11, it is hard to look at the graphs going back to Figure 6 where the strain gauge location was reported. It is suggested to improve these figures by adding at their side a small drawing with the indication of those strain gauges whose strain is plotted in the curves.
Moreover. Figures 10 and 11 report the Load value on the vertical axis: adding a second vertical axis on the right with the corresponding stress values on the steel elements would be better.
Finally, in the same figures, the small zoom boxes added within the main graph keep being too small and unreadable: they should be resized to improve readability.
4.1 Numerical simulation of composite girder
Is the mesh size indicated in Figure 14 the same for every model part, i.e. 50 mm? Did you perform any sensitivity analysis to optimize the mesh type and size with respect to the computational time required by the analysis? Can you provide information about the computational time?
Can you clarify the following sentence? “The steel-concrete interlayer interaction was simulated using the “General Contact” command, while the tangential and normal direction was respectively simulated by penalty frictional formulation and ‘Hard’ contact.” Which are the mechanical features of the contact implemented in the so-called “General contact”? Where did you use the general contact, and where the tangential and normal contact mentioned in your sentence? Please, clarify.
Can you also clarify the following sentence? “The loading scheme adopted “uniform” loading, while the displacement boundary conditions took linear constraint support”. You should clarify what kind of loading scheme you are applying (static load, monotonic load, etc.) and if you are loading the surface loaded in the real specimen during your test or just one local point in the midspan of the beam, as it looks you’re your Figure 14. Then, what do you mean for “linear constraint support”? Are you modelling the supports with a sort of linear springs? Or what else? Please, clarify.
4.2 Material properties
Can you specify what kind of implementation you have adopted for the tensile behaviour of concrete? Did you adopt the stress-strain method or others? Have you tried to check if you get any difference in the results of the analysis or in its convergence if you change the concrete tensile model? How did you implement a different elastic modulus in tension and compression for concrete?
Finally, you should add a comment on how you evaluate the compressive and tensile damage factors of the concrete, dc and dt.
References
The reference list should be increased with the reference studies indicated before. Moreover, the reference to the finite element code Abaqus should be added.
Author Response
We sincerely appreciate the valuable comments, suggestions and questions from the reviewer. The manuscript has been revised accordingly, in which the changes have been highlighted for the review convenience. Meanwhile, the point-to-point responses to the reviewer's comments are listed as follows.
- Introduction:
The introduction should be proposed according to a more holistic approach. To this scope, the authors should mention that the analysis of steel-to-concrete connections in hybrid beams (stress transfer mechanisms, seismic response, etc.) refers not only to the beams and applications mentioned in their paper but also to classical composite beams, box beams and non-standard composite beams such as hybrid trussed beams. The latter, usually adopted for civil structures and infrastructures (e.g., https://doi.org/10.1016/j.engstruct.2012.11.004), in the form of full- and small-thick geometry (e.g., https://doi.org/10.1016/j.conbuildmat.2017.11.134) with different failure modes. These girders have been studied in recent years and analyzed by several authors through detailed finite element contact models and theoretical formulations that evidence how different steel-concrete interactions can determine different failure modes in the beam. Also, the size effect on the structural performance should be mentioned, which is typically evaluated through numerical models when laboratory tests cannot be performed. Finally, the seismic damping capacity should be treated (e.g., http://ingegneriasismica.org/dissipative-connections-of-rc-frames-with-prefabricated-steel-trussed-concrete-beams)
Response: We thank the reviewer for helping us to enlarge the state of the art with a more holistic approach of the steel to concrete connections. We have improved the introduction part according to the reviewer’s comments. References about the analysis of steel-to-concrete connections and seismic damping capacity were added. Please refer to Section 1 in the revised manuscript.
- Specimen design:
with reference to Figures 3a) and b), please improve the readability of the quotes of holes.
Response: We have improved the readability of the quotes of holes. Please refer to Section 2.1. in the revised manuscript.
- Loading and monitoring setup:
With reference to Figure 6a), maybe it would be better to improve the indicators of the Strain gauge arranged at quarter-span because it is not actually placed at the quarter-span in your drawing.
Response: We have adjusted the position in Figure 6a). Please refer to Section 2.3 in the revised manuscript.
- Strains in steel flange and reinforcing bars:
With reference to Figure 10 and Figure 11, it is hard to look at the graphs going back to Figure 6 where the strain gauge location was reported. It is suggested to improve these figures by adding at their side a small drawing with the indication of those strain gauges whose strain is plotted in the curves.
Moreover. Figures 10 and 11 report the Load value on the vertical axis: adding a second vertical axis on the right with the corresponding stress values on the steel elements would be better.
Finally, in the same figures, the small zoom boxes added within the main graph keep being too small and unreadable: they should be resized to improve readability.
Response: We have added the strain gauge position to Figure 10 and Figure 11 and also improved the partial enlarged drawing. However, the strain value cannot be directly converted to stress value by multiplying the elasticity modulus in the test result. That is because the elasticity modulus of the test specimen varies during the loading after the elastic stage. Besides, the local buckling also contributed to an increase of strain, while the stress value didn’t increase accordingly.
- Numerical simulation of composite girder:
Is the mesh size indicated in Figure 14 the same for every model part, i.e. 50 mm? Did you perform any sensitivity analysis to optimize the mesh type and size with respect to the computational time required by the analysis? Can you provide information about the computational time?
Can you clarify the following sentence? “The steel-concrete interlayer interaction was simulated using the “General Contact” command, while the tangential and normal direction was respectively simulated by penalty frictional formulation and ‘Hard’ contact.” Which are the mechanical features of the contact implemented in the so-called “General contact”? Where did you use the general contact, and where the tangential and normal contact mentioned in your sentence? Please, clarify.
Can you also clarify the following sentence? “The loading scheme adopted “uniform” loading, while the displacement boundary conditions took linear constraint support”. You should clarify what kind of loading scheme you are applying (static load, monotonic load, etc.) and if you are loading the surface loaded in the real specimen during your test or just one local point in the midspan of the beam, as it looks you’re your Figure 14. Then, what do you mean for “linear constraint support”? Are you modelling the supports with a sort of linear springs? Or what else? Please, clarify.
Response: We have added a sensitivity analysis in section 4.1, which specifically illustrates the determination of each parameter during modeling. The calculation can be completed within 50 minutes using a 64-core workstation. We have added the explanation in section 4.1.
“General Contact” is a commend from Abaqus which contains a series of contact parameters (including penalty frictional formulation and ‘Hard’ contact). In this simulation, all the contacts between the steel and concrete were simulated by penalty frictional formulation along tangential direction and ‘Hard’ contact along normal direction. We have added explanations in section 4.1 to clarify the contact condition.
“Uniform loading” is also a commend in Abaqus, which evenly distributed the load on the loading surface. Thus, it is according to the actual test loading condition. Linear constraint support indicated that all the elements’ nodes along the constraint line were restricted. We have added explanation in section 4.1 and modified Figure 14 to clarify the loading and boundary conditions.
- Material properties:
Can you specify what kind of implementation you have adopted for the tensile behaviour of concrete? Did you adopt the stress-strain method or others? Have you tried to check if you get any difference in the results of the analysis or in its convergence if you change the concrete tensile model? How did you implement a different elastic modulus in tension and compression for concrete?
Finally, you should add a comment on how you evaluate the compressive and tensile damage factors of the concrete, dc and dt.
Response: We adopted stress-strain method during simulation. Comparison between different concrete constitutive models was added in the sensitive analysis in Section 4.1. Actually, the concrete constitutive model provided by the Eurocode should pay attention to parameters such as the fracture energy and characteristic length for convergence. The concrete constitutive model used in this paper is provided by Chinese Standard GB50010-2010 (Code for design of concrete structures), which adopted empirical parameters to determine the descending part of the concrete constitutive curve. Only some parameters in the equation (tensile and compressive elastic strains) took reference to the Eurocode. To clarify the material properties, we have added specific explanations of the model in section 4.2.
- References:
The reference list should be increased with the reference studies indicated before. Moreover, the reference to the finite element code Abaqus should be added.
Response: We have increased the reference list according to the reviewer’s suggestion and included ABAQUS in the reference list. Please refer to Section 1 and Section 4.1 in the revised manuscript.
Reviewer 4 Report
The intensive references are presented. The references show that it is one more contribution into the field. The field is practically important, so the paper may be estimated a once more discussion on the problem.
Questions:
No doubt, between the concrete and steel parts, there must exist a connection zone. Is it necessary, a zone made of composite material? Composite is not forbidden for this purpose. There were published papers on the general theory of joined elastic bodies.
It seems the composite is treated as a homogeneous material in the paper. Which specific features of composite are essentially used to solve this practical problem?
Author Response
We sincerely appreciate the valuable comments, suggestions and questions from the reviewer. The point-to-point responses to the reviewer's comments are listed as follows.
- No doubt, between the concrete and steel parts, there must exist a connection zone. Is it necessary, a zone made of composite material? Composite is not forbidden for this purpose. There were published papers on the general theory of joined elastic bodies.
Response: Hybrid girder bridge includes steel segment and concrete segment. There is great deviation between the section dimension of the two segments, while stress concentration occurs if the steel segment is connected directly with the concrete segment. The common method to connect the steel and concrete segment is to adopt a composite segment which can integrate well with both segments of different materials and create an smooth transition of stresses between the steel and the concrete parts of the hybrid girder bridge, avoiding stress concentration in the concrete that could damage/break it.
- It seems the composite is treated as a homogeneous material in the paper. Which specific features of composite are essentially used to solve this practical problem?
Response: Actually, the steel-concrete joint in the hybrid girder adopted both steel material and concrete material. In the test specimen, the steel girder was firstly fabricated and then concrete was casted into the steel girder. The steel-concrete joint is commonly adopted in hybrid bridges because it can integrate well with both steel segment and concrete segment and provide appropriate stress distribution in both materials. The specific features of composite to solve the problem of the hybrid girder bridge are those of general applicability to the steel-concrete composite structures: the good connection between steel and concrete and the concrete and steel helping each other where the problems can arise: the steel placed on the zones of high tensile stress (that concrete can not resist) and the concrete placed in the zones with high compression stress, avoiding the bucking of the steel.
Round 2
Reviewer 2 Report
As there are some limitations mentioned in the paper, please rewrite the objective in Line 89-91.
Author Response
Thanks to the reviewer for this comment. The objective has been re-written taking into account the limitations mentioned in the paper.